# Frequency and Genetic Analysis of Porcine Circovirus Type 2, Which Circulated between 2014 and 2021 in Jiangsu, China

**DOI:** 10.3390/ani14192882

**Published:** 2024-10-07

**Authors:** Qi Xiao, Meng Qu, Jianping Xie, Cigen Zhu, Yuping Shan, Aihua Mao, Wenxian Qian, Jiaping Zhu, Jiahui Guo, Dong Lang, Jiaqiang Niu, Libin Wen, Kongwang He

**Affiliations:** 1Institute of Veterinary Medicine, Jiangsu Academy of Agricultural Sciences, Nanjing 210014, China; xiaoqi-2122@163.com (Q.X.); qumeng@ysfri.ac.cn (M.Q.); 20100997@jaas.ac.cn (J.X.); mah8680@126.com (A.M.); kim_qwx@163.com (W.Q.); 13949119396@163.com (J.Z.); guojiaqian927@163.com (J.G.); 2022807118@stu.njau.edu.cn (D.L.); 2Key Laboratory of Veterinary Biological Engineering and Technology, Ministry of Agriculture and Rural Affairs, Nanjing 210014, China; 3Jiangsu Co-Innovation Center for Prevention and Control of Important Animal Infections Diseases and Zoonoses, Yangzhou University, Yangzhou 225009, China; 4Jiangsu Key Laboratory for Food Quality and Safety-State Key Laboratory Cultivation Base of Ministry of Science and Technology, Nanjing 210014, China; 5Jiangsu Animal Husbandry Station, Nanjing 210036, China; zhucigen@gmail.com; 6Animal Husbandry and Veterinary Station of Lianyungang, Lianyungang 222000, China; 13961390836@163.com; 7College of Animal Science, Tibet Agricultural and Animal Husbandry University, Provincial Key Laboratory of Tibet Plateau Animal Epidemic Disease Research, Linzhi 860000, China; lznjq@163.com

**Keywords:** porcine circovirus type 2, Jiangsu Province, genetic analysis, genotype

## Abstract

**Simple Summary:**

Porcine circovirus type 2 (PCV2) is currently the smallest known animal virus, and the diseases caused by its infection have caused huge economic losses to the pig farming industry worldwide. The virus evolves rapidly and is divided into nine genotypes, some of which pose significant challenges in preventing and controlling the disease. In this study, the polymerase chain reaction technique and bioinformatic analysis were used to investigate PCV2 infection in pig farms in Jiangsu Province, China. The results showed that the infection rate of PCV2 in pig farms in Jiangsu Province has been increasing annually, with genotypes PCV2a, PCV2b, PCV2d, and PCV2i. The genomic characteristics of the recently confirmed genotype PCV2i were elucidated.

**Abstract:**

Porcine circovirus-associated diseases, caused by porcine circovirus type 2 (PCV2), are widespread and result in significant economic losses to the global swine industry. PCV2 can currently be divided into nine genotypes (PCV2a to PCV2i), with the currently dominant one being the PCV2d genotype. In this study, 2675 samples from 804 pig farms in 13 cities in Jiangsu Province, China, were collected between 2014 and 2021 and subjected to polymerase chain reaction analysis to investigate the frequency and genetic diversity of PCV2. The results showed that 41.42% (1108/2675) of samples tested positive for PCV2. The researchers further analyzed the genetic characteristics of 251 PCV2 strains and found that they belonged to the following four genotypes: PCV2a, PCV2b, PCV2d, and PCV2i. The dominant genotype was PCV2d, with a frequency of 49.80% (125/251). The detection rate of PCV2b was significantly higher than those of PCV2a and PCV2i, at 35.46% (89/251), 7.57% (19/251), and 7.17% (18/251), respectively. The percentage of different genotypes of PCV2 varied irregularly over time. We have further revealed the fingerprint of PCV2i genomic nucleotides for the first time. In conclusion, this study illustrates the high frequency and evolutionary features of PCV2 in Jiangsu Province over the past few years.

## 1. Introduction

Porcine circoviruses (PCVs) are the smallest, circular, single-stranded DNA viruses belonging to the genus Circovirus of the family Circoviridae. The following four species of PCVs with similar structures have been identified: porcine circovirus type 1 (PCV1), PCV2, PCV3, and PCV4 [1,2,3,4,5]. In addition, other species of PCVs, including porcine circovirus-like viruses and porcine circovirus-like mini agents, have been reported [6]. PCV2 has been considered the primary causative agent of porcine circovirus-associated diseases (PCVADs) since its identification in Canada in the 1990s. Post-weaning multisystemic wasting syndrome (PMWS) is the most common PCVAD in clinical practice, characterized by wasting, enlargement of lymph nodes, skin pallor, and dyspnea in fattening pigs, and its histopathological lesions include lymphocyte depletion with granulomatous inflammation of lymphoid tissues, and interstitial pneumonia, which causes serious economic losses to the pig industry worldwide [7,8,9].

The genome of PCV2 is approximately 1767 nucleotides (nt) in length and comprises eleven open reading frames (ORFs), of which six ORFs have been identified [10,11]. Replication-related proteins (Rep and Rep’) are encoded by ORF1, whereas the capsid protein (Cap) is encoded by ORF2 [12,13,14]. Compared with other ORFs, the ORF2 gene can be used for genetic evolution analysis due to its high mutation rate. A nucleotide diversity (p-distance) cutoff value of 3.5% (for ORF2) or 2.0% (for the complete genome) was used to distinguish between different PCV2 genotypes [15]. Nine genotypes (PCV2a–PCV2i) have been identified [16,17]. PCV2 strains circulating in pigs before 2008 belonged to three genotypes (PCV2a–PCV2c). PCV2d was identified in 2010 and has become prevalent worldwide since 2012 [18]. Other novel PCV2 genotypes (PCV2e–PCV2i) have been identified in recent years [16,17,19,20,21]. Generally, there are two changes in the dominant genotype of PCV2: one from PCV2a to PCV2b and another from PCV2b to PCV2d [22,23]. It is interesting that the genotype shifts of PCV2 seem to enhance the viral virulence, although there are also different reports [24,25,26]. According to previous reports, the genotype shifts and emerging genotypes of PCV2 are related to factors such as immune selective pressure derived from vaccination or natural infection, international trades, and viral evolution [27,28,29].

Previous studies have examined the occurrence of PCV in Jiangsu Province, and the results showed that the positive rate of PCV2 in eastern China, including Jiangsu Province, was 35.33% [30,31]. However, there is a lack of comprehensive data on the genetic variety and frequency of PCV2 in Jiangsu pig farms. Therefore, an investigation was conducted from 2014 to 2021 using polymerase chain reaction (PCR) to determine the epidemiological characteristics of PCV2 in Jiangsu Province. In addition, the complete genomes of 251 PCV2 strains from various periods and cities were analyzed.

## 2. Materials and Methods

### 2.1. Clinical Sample Collection

From April 2014 to September 2021, 2675 clinical samples (lung, serum, diarrheal stool, and lymph node) were collected from 804 indoor pig farms in the following 13 cities in Jiangsu Province: Nanjing, Suzhou, Wuxi, Changzhou, Zhenjiang, Nantong, Huaian, Yangzhou, Taizhou, Yancheng, Suqian, Lianyungang, and Xuzhou. Most of the pigs from these farms showed clinical signs of PMWS, and the samples were randomly collected from pigs with PMWS and clinically healthy pigs (Table 1). All pigs were vaccinated with the commercial PCV2 vaccines.

### 2.2. DNA Extraction and PCV2 Detection

Commercial kits (TIANDZ, Bejing, China) were used to extract viral DNA from the samples, according to the manufacturer’s instructions. Subsequently, the presence of PCV2 nucleic acid was detected using PCR with the previously published primers (F: 5′-TAGACGGATATTGTAGTCC-3′ and R: 5′-TTCCGCAGAAGAAGACAC-3′) [32]. PCR conditions were as follows: 95 °C for 5 min; 35 cycles of 94 °C for 30 s; 58 °C for 30 s; 72 °C for 30 s; and a final extension for 10 min at 72 °C. The PCR products were analyzed using 1.5% agarose gel electrophoresis, in which the samples with the expected 630 bp DNA band were recognized as PCV2-positive samples.

### 2.3. Complete Genome Sequencing of PCV2

According to the collected cities and years of the positive samples, 251 PCV2-positive samples were randomly selected to amplify the complete genome of the virus, as described previously [33]. PCR amplification was performed using the following steps: 95 °C for 5 min; 35 cycles of 94 °C for 30 s; 58 °C for 30 s; and 72 °C for 90 s, followed by 72 °C for 10 min. The PCR products were subsequently purified using a Gel Extraction Kit (AXYGEN, Hangzhou, China), cloned into the pMD18-T vector (TaKaRa Biotechnology Co., Ltd., Dalian, China), and sequenced by Genscript Biotech Co., Ltd. (Nanjing, China).

The sequences of the 251 novel PCV2 strains were submitted to GenBank (OR533426–OR533483, OR537631–OR537693, OR542932–OR542954, OR553617–OR553646, and OR567166–567242).

### 2.4. Bioinformatics Analyses

The complete genomes of the following eight reference PCV2 strains were downloaded from GenBank: PCV2a (AF381175), PCV2b (AY181945), PCV2c (EU148503), PCV2d (AY181946), PCV2e (KT795289), PCV2f (MF278779), PCV2g (JX099786), and PCV2h (JX506730) [16,34]. The Lasergene DNAStar software (7.1) was used to analyze the nucleotide sequences (complete, ORF1, and ORF2) and their corresponding amino acid (aa) sequences (Cap and Rep) of 251 novel PCV2 strains and reference strains. Phylogenetic trees were generated based on the complete genome sequences using the neighbor-joining method with maximum composite likelihood model in MEGA 7.0 software with 1000 bootstrap replications.

### 2.5. Frequency Data Analyses

The frequency data analysis of PCV2 in pigs in Jiangsu Province was performed using the chi-square test with Excel 2010, including infection rates by year and rates in different cities. Statistical significance was set at *p* < 0.05.

## 3. Results

### 3.1. Frequency of PCV2 in Jiangsu Province from 2014 to 2021

In this study, 2675 samples from 804 pig farms in 13 cities in Jiangsu Province were collected for PCV2 nucleic acid analysis using conventional PCR. The results showed that the positive rate of PCV2 was 41.42% (1108/2675), with positive rates varying from 28.87 to 63.64% in different cities and from 27.89 to 63.33% in different years. In 2021, the infection rate of PCV2 in Jiangsu Province was the highest at 63.33% (95/150), and the infection rate was the lowest in 2018 at 27.89% (94/337). From a regional perspective, the infection rate of PCV2 was the highest in Zhenjiang, at 63.64% (42/66), and the infection rate was the lowest in Changzhou, at 28.87% (28/97) (Table 2). There was an extremely significant difference in the infection rate of PCV2, both in terms of year and city (*p* < 0.01). At the farm level, the positivity rate of PCV2 ranged from 37.14 to 71.43% in different cities and from 33.33 to 91.53% in different years, averaging 55.72% (448/804) (Table 3).

### 3.2. Genome Sequence Analysis

In this study, 251 PCV2 strains were randomly selected from different regions among the PCV2-positive samples to investigate the genetic features of recent PCV2 strains prevalent in Jiangsu Province (Table 2). The complete genome sequences of all 251 PCV2 strains were amplified, sequenced, and analyzed and were found to be 1766–1768 nt in length. The ORF1 gene encoding Rep and ORF2 gene encoding Cap had lengths of 945 nt and 702–705 nt, respectively. The 251 PCV2 strains shared 94.1–100.0% (complete genome), 96.1–100.0% (ORF1), and 89.5–100.0% (ORF2) identity at the nt level and 96.5–100.0% (Rep protein) and 85.9–100.0% (Cap protein) identity at the aa level. The PCV2 strains shared 100% complete genome nucleotide identity, coming from either one farm or different farms. Interestingly, a nucleotide deletion at site 1039 nt was unique to the JS890 strain, which caused a change in the amino acid sequence of the Cap, from 233PK to 233LSE. In contrast, sites 1356 (nucleotide deletion) and 1389 (nucleotide insertion) were unique to the JS953 strain, resulting in a change of amino acid sequence of the Cap, from 118VGSSAVILDD127 to 118SGLQCCYSRY127. These amino acid changes in the capsid protein caused by base deletions or insertions may affect the epitopes of the virus [35].

### 3.3. Phylogenetic Analysis

Phylogenetic trees were generated, based on the complete genome sequences of 251 PCV2 strains identified in this study. The 251 PCV2 strains were divided into the following four genotypes: PCV2a, PCV2b, PCV2d, and PCV2i. Phylogenetic tree analysis further revealed that PCV2a and PCV2i strains had a closer phylogenetic relationship than PCV2b and PCV2d. Among the 251 strains, 19 (7.57%) belonged to genotype 2a, 18 (7.17%) to genotype 2i, 89 (35.46%) to genotype 2b, and nearly half of the strains (125) to genotype 2d (Figure 1).

The positive rate of different genotypes of PCV2 in Jiangsu Province varied with different years and samples, such as the following: the PCV2d-positive rate was 1.23% (5/408), 7.78% (43/553), 1.74% (8/459), 8.42% (39/463), 3.56% (12/337), 4.12% (7/170), 3.70% (5/135), and 4.00% (6150/) in 2014, 2015, 2016, 2017, 2018, 2019, 2020, and 2021, respectively, and PCV2d-positive sample rate was 6.14% (106/1725) in lung, 0.00% (0/427) in serum, 4.33% (13/300) in diarrheal stool, and 2.69% (6/223) in lymph node (Table 4).

Although the frequency of PCV2d fluctuated, PCV2d was generally the dominant genotype that circulated between 2014 and 2021, followed by PCV2b, whereas the frequency of PCV2i was lower than that of PCV2b. There were two PCV2i strains in Xuzhou and one PCV2i strain in Yancheng in 2014, four strains in Nantong and one strain in Yancheng in 2015, one strain in Huaian and one strain in Lianyungang in 2016, one strain in Changzhou and one strain in Lianyungang in 2017, one strain in Nanjing, one strain in Suqian, and one strain in Yancheng in 2018, two strains in Nanjing in 2019, and one strain in Huaian in 2021. Notably, PCV2a still has a low frequency (Figure 2).

The sequence similarity range of the novel PCV2a strains was 98.9–100.0% (complete genome), 99.2–100.0% (ORF1), and 98.3–100.0% (ORF2) at the nt level and 99.4–100.0% (Rep protein) and 98.3–100.0% (Cap protein) at the aa level.

In contrast, sequence similarities among the novel PCV2b strains varied between 98.0 and 100.0% (complete genome), 98.1 and 100.0% (ORF1), and 97.5 and 100.0% (ORF2) at the nt level and 98.1 and 100.0% (Rep protein) and 95.6 and 100.0% (Cap protein) at the aa level.

The sequence similarity among the novel PCV2d strains was 96.2–100.0% (complete genome), 96.5–100.0% (ORF1), and 95.3–100.0% (ORF2) at the nt level, and the variation at the aa level was 97.1–100.0% (Rep protein) and 94.7–100.0% (Cap protein).

Finally, pairwise sequence comparisons among the novel PCV2i strains varied between 98.5 and 100.0% (complete genome), 98.7 and 100.0% (ORF1), and 98.1 and 100.0% (ORF2) at the nt level and 98.1 and 100.0% (Rep protein) and 97.9 and 100.0% (Cap protein) variation at the aa level.

### 3.4. Analysis of Genomic Sequences of PCV2i Strains

In a recent study, a new genotype called PCV2i was proposed through phylogenetic analysis, but not much information is available about this genotype [17]. To investigate the sequence characteristics of PCV2i obtained in this study, a sequence alignment was performed among the 251 PCV2 strains. The complete genome of all 18 PCV2i strains was 1768 nt in length, and the length of the Cap sequence was 233 aa.

Based on the complete genome sequence, the nucleotide sequence identity between PCV2i and PCV2a was 96.0–97.3%, 94.8–96.0% (PCV2b), and 93.8–95.6% (PCV2d), respectively, and 93.3–94.7% (PCV2a), 91.1–92.5% (PCV2b), and 91.6–92.9% (PCV2d), respectively, based on the ORF2 nucleotide sequence.

For the 251 PCV2 strains and eight reference strains in this study, nucleotides at positions 617 (T), 1164 (T), 1596 (C), and 1604 (A) were exclusively present in PCV2i strains. Additionally, aa residues 47 (S) and 191 (K) in the Cap of PCV2i strains were identified as potential signature markers for PCV2i, and the latter might affect the epitope of PCV2, as residue 191 (K) was located at antigenic epitope 175–192 [35] (Table 5).

In addition, the nucleotide sites 1175, 1177, 1226, 1313, 1343, and 1673 of all PCV2i strains were A, T, G, G, A, and T, respectively, which are currently only observed in PCV2f with A at position 1175, in PCV2a with T at 1177, in PCV2f with G at position 1226, in PCV2f with G at position 1313, in PCV2f with A at position 1343, and in PCV2e with T at position 1673.

## 4. Discussion

The disease caused by PCV2, known as PCVAD, is a major threat to the pig industry worldwide, as it is considered a global epidemic [7,36,37,38]. The first report of PCV2 infection in China was published in 2000; since then, it has become widespread throughout the country [39].

To study the epidemiological characteristics of PCV2 in Jiangsu Province, southern China, 2675 pig samples were collected. Previous research has shown that the infection rates of PCV2 in breeding farms in southern China are higher than those in northern China [40]. However, the PCV2-positive rate among the samples collected in Jiangsu Province was 42.58%, which is similar to that in Shandong Province (36.98%, 490/1325) [41] in northern China but lower than that in Henan (62.4%, 73/117) in northern China [42].

PCV2 has a higher evolutionary rate than other DNA viruses, with a rate of 10-3–10-4 substitutions/site/year, which is similar to that of RNA viruses [43]. In the past, the PCR–RFLP method was employed for PCV2 genotyping, which used nucleotide sequence amplification and endonuclease digestion. Research conducted in Canada, Japan, and China using this method showed that there were five RFLP types (A–E) of PCV2 in Canada, at least five RFLP types in Japan, and nine different genotypes (A–I) in China [44,45,46].

To distinguish between different PCV2 genotypes, pairwise sequence comparison (PASC) analysis, which is based on the proportion of nucleotide sites at which two sequences differ (p-distance), is widely used. A diversity cutoff of 2.0% for the complete genome and 3.5% for ORF2 has been proposed [15].

A new PCV2 genotyping methodology based on three criteria was proposed by Franzo and Segalés in 2018. The first criterion was a maximum intragenotype p-distance of 13%, which was calculated for the ORF2 gene. The second criterion was bootstrap support at a corresponding internal node higher than 70%, and the third criterion was at least 15 available sequences. Based on these criteria, PCV2 has been divided into eight genotypes ranging from PCV-2a to PCV-2h [16]. Subsequently, a new genotype, PCV2i, was proposed through phylogenetic analysis [17].

PCV2 has undergone two significant global genotypic shifts. The dominant PCV2a genotype was replaced with PCV2b from 2002 to 2008. Since 2009, PCV2d has replaced PCV2b as the dominant genotype worldwide [47,48]. PCV2d is considered to be more virulent than PCV2a and PCV2b, with severe clinical signs, viremia, and lesions [24]. However, there are also reports suggesting that the virulence of PCV2d is similar to that of PCV2a and PCV2b [49,50]. Notably, the coprevalence of multiple PCV2 genotypes or with other circoviruses, such as PCV3, PCV4, and PCV-like virus P1, often appears in a pig farm or even in a sample [51,52,53].

Only the following four genotypes were identified among the 251 PCV2 strains obtained in this study: PCV2a, PCV2b, PCV2d, and PCV2i, with proportions of 7.57, 35.46, 49.80, and 7.17%, respectively. This indicates that PCV2b and PCV2d are the predominant genotypes in Jiangsu Province. Specifically, PCV2b and PCV2d were the dominant genotypes circulating in the Jiangsu Province from 2015 to 2017. Although the detection rates of PCV2a and PCV2i were similar, PCV2a was not detected before 2017, whereas PCV2i was scattered in samples from 2014 to 2021.

This study provides evidence supporting the existence of PCV2i as a new genotype via PASC analysis, and Franzo and Segalés proposed a new genotyping methodology or phylogenetic analysis. However, its genomic characteristics have not been elucidated since it has been recommended as a new genotype. This study revealed four characteristic nucleotide changes in the PCV2i genome at sites 617 (T), 1164 (T), 1596 (C), and 1604 (A), which resulted in characteristic changes in aa residues 47 (S) and 191 (K) of the viral capsid protein compared with those of other genotypes.

PCV2 Cap is the only structural protein involved in multiple biological processes, including viral invasion, replication, and immune response [54]. Studies have shown that certain aa residues in the protein are critical for viral nuclear localization and epitope recognition. Specifically, residues 12–18 and 34–41 are important for nuclear localization [55], while residues 47–63, 165–200, and the last four aa residues at the C-terminus are crucial for epitope recognition [35,56]. Therefore, the two aa mutations in the Cap of PCV2i may determine the biological characteristics of the virus. Further research is needed to determine the virulence of PCV2i and the effectiveness of current vaccines against it. Interestingly, mutations in specific nucleotides of PCV2i have led to new endonuclease digestion sites that can be used for PCR-RFLP genotyping of PCV2.

## 5. Conclusions

This study sheds light on the epidemiology and genetic characteristics of PCV2 that circulated between 2014 and 2021 in Jiangsu Province, China. These findings reveal a high frequency of multiple PCV2 genotypes in the region and provide insights into the genomic characteristics of PCV2i.

## Figures and Tables

**Figure 1 animals-14-02882-f001:**
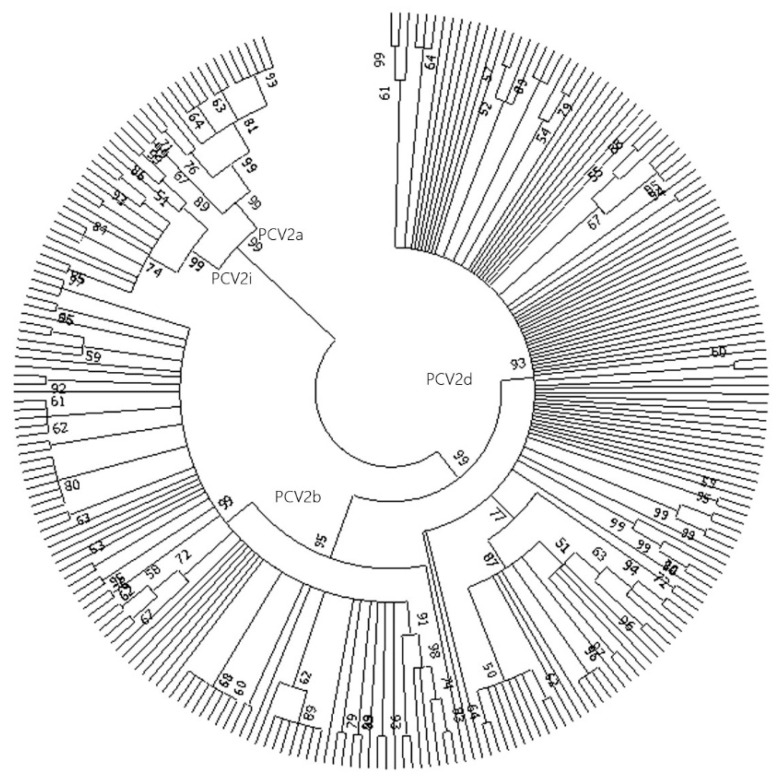
Phylogenetic tree based on 251 PCV2 complete sequences used in this study. The tree was constructed with MEGA7.0 software using the neighbor-joining (NJ) method with1000 bootstrap replications. The representative strains of different genotypes of PCV2 used in this study are PCV2a (AF381175), PCV2b (AY181945), PCV2c (EU148503), PCV2d (AY181946), PCV2e (KT795289), PCV2f (MF278779), PCV2g (JX099786), and PCV2h (JX506730).

**Figure 2 animals-14-02882-f002:**
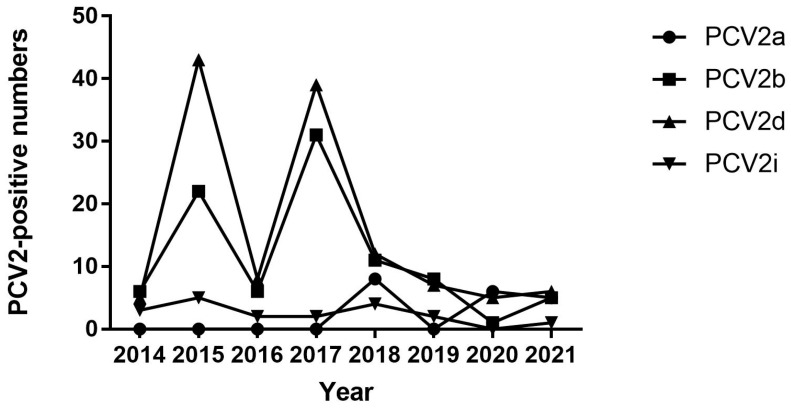
Proportion of different PCV2 genotypes in 251 strains in Jiangsu from 2014 to 2021.

**Table 1 animals-14-02882-t001:** Information of samples obtained from 2014 to 2021 in Jiangsu Province, China.

Collection Year	Sample Type	Sample Number
2014	lung, serum, diarrheal stool, and lymph node	408
2015	lung, diarrheal stool, and lymph node	553
2016	lung, diarrheal stool, and lymph node	459
2017	lung, serum, diarrheal stool, and lymph node	463
2018	lung	337
2019	lung, and lymph node	170
2020	lung, diarrheal stool	135
2021	lung	150

**Table 2 animals-14-02882-t002:** Infection rate of PCV2 in 13 cities in Jiangsu Province from 2014 to 2021.

	2014	2015	2016	2017	2018	2019	2020	2021
Changzhou	0/10 (0)	10/41 (2)	9/19 (2)	0/5 (0)	3/7 (3)	4/9 (4)	0/2 (0)	2/4 (2)
Huaian	8/30 (2)	18/60 (3)	15/10 (2)	35/113 (4)	8/52 (2)	32/48 (4)	4/11 (4)	17/25 (2)
Lianyungang	7/12 (2)	27/41 (4)	18/55 (3)	22/45 (3)	7/31 (2)	15/21 (3)	9/19 (2)	20/32 (3)
Nanjing	30/48 (4)	4/13 (4)	8/17 (2)	6/20 (2)	3/4 (3)	6/10 (2)	4/7 (4)	6/9 (2)
Nantong	40/74 (5)	21/78 (3)	10/28 (2)	19/29 (3)	8/32 (2)	7/8 (2)	9/18 (2)	9/16 (2)
Suzhou	12/23 (3)	10/32 (2)	11/28 (2)	2/3 (2)	0/1 (0)	4/20 (4)	0/0 (0)	2/5 (2)
Taizhou	20/59 (3)	17/47 (3)	16/22 (3)	28/41 (4)	9/16 (2)	2/5 (2)	8/20 (2)	4/6 (4)
Wuxi	13/13 (3)	17/32 (3)	8/9 (2)	0/5 (0)	2/3 (2)	0/3 (0)	2/5 (2)	1/3 (1)
Suqian	13/18 (3)	29/72 (4)	18/95 (3)	10/22 (2)	14/36 (3)	22/24 (4)	10/16 (2)	12/15 (2)
Xuzhou	7/35 (2)	15/42 (3)	7/16 (2)	28/66 (4)	22/109 (4)	3/11 (3)	7/16 (2)	3/5 (3)
Yancheng	8/46 (2)	35/56 (5)	7/35 (2)	24/72 (3)	10/22 (2)	9/9 (2)	5/12 (5)	10/18 (2)
Yangzhou	16/21 (3)	12/18 (2)	13/23 (2)	7/35 (2)	6/18 (2)	2/2 (2)	6/7 (2)	8/10 (2)
Zhenjiang	12/19 (3)	10/21 (2)	9/9 (2)	7/7 (2)	2/6 (2)	0/0 (0)	1/2 (1)	1/2 (1)

Note: The numbers in parentheses represent the number of sequenced PCV2 strains.

**Table 3 animals-14-02882-t003:** PCV2 positivity rate in different pig farms without double counting in 13 cities in Jiangsu Province from 2014 to 2021.

	2014	2015	2016	2017	2018	2019	2020	2021
Changzhou	0.00% (0/4)	42.86% (6/14)	33.33% (3/9)	0.00% (0/1)	100.00% (2/2)	33.33% (1/3)	0.00% (0/1)	100.00% (1/1)
Huaian	50.00% (4/8)	35.00% (7/20)	26.09% (6/23)	55.00% (11/20)	27.27% (3/11)	90.00% (9/10)	60.00% (3/5)	90.00% (9/10)
Lianyungang	100.00% (4/4)	71.43% (10/14)	53.85% (7/13)	64.71% (11/17)	30.00% (3/10)	83.33% (5/6)	55.56% (5/9)	100.00% (10/10)
Nanjing	66.67% (6/9)	28.57% (2/7)	37.50% (3/8)	42.86% (3/7)	100.00% (1/1)	75.00% (3/4)	50.00% (1/2)	100.00% (5/5)
Nantong	57.14% (8/14)	60.00% (12/20)	30.77% (4/13)	45.45% (5/11)	45.45% (5/11)	100.00% (4/4)	80.00% (4/5)	83.33% (5/6)
Suzhou	60.00% (3/5)	44.44% (4/9)	33.33% (2/6)	100.00% (1/1)	0/00% (0/1)	50.00% (3/6)	0.00% (0/0)	100.00% (2/2)
Taizhou	61.54% (8/13)	47.37% (9/19)	50.00% (4/8)	81.82% (9/11)	62.50% (5/8)	100.00% (2/2)	50.00% (4/80	50.00% (1/2)
Wuxi	100.00% (3/3)	63.64% (7/11)	40.00% (2/5)	0.00% (0/2)	100.00% (1/1)	0/00% (0/1)	50.00% (1/2)	50.00% (1/2)
Suqian	57.14% (4/7)	52.38% (11/21)	17.39% (4/23)	62/50% (5/8)	25.00% (3/12)	100.00% (8/8)	100.00% (5/5)	100.00% (7/7)
Xuzhou	44.44% (4/9)	60.00% (9/15)	42.86% (3/7)	61.54% (8/13)	20.00% (4/20)	40.00% (2/5)	66.67% (4/6)	50.00% (1/2)
Yancheng	71.43% (5/7)	77.27% (17/22)	20.00% (3/15)	60.00% (12/20)	50.00% (3/6)	100.00% (4/4)	50.00% (2/4)	100.00% (8/8)
Yangzhou	66.67% (6/9)	87.50% (7/8)	60.00% (3/5)	44.44% (4/9)	60.00% (3/5)	100.00% (1/10	100.00% (2/2)	100.00% (3/3)
Zhenjiang	71.43% (5/7)	58.33% (7/12)	66.67% (2/3)	100.00% (3/3)	100.00% (1/1)	0.00% (0/0)	100.00% (1/1)	100.00% (1/1)

**Table 4 animals-14-02882-t004:** Distribution of PCV2 genotypes in different samples from pig farms in Jiangsu Province from 2014 to 2021.

		Lung	Serum	Diarrheal Stool	Lymph Node
2014	PCV2a	0 (202)	0 (132)	0 (48)	0 (26)
	PCV2b	5	1	0	0
	PCV2d	3	0	2	0
	PCV2i	3	0	0	0
2015	PCV2a	0 (342)	0 (114)	0 (37)	0 (60)
	PCV2b	22	0	0	0
	PCV2d	41	0	1	1
	PCV2i	5	0	0	0
2016	PCV2a	0 (219)	0 (36)	0 (140)	0 (64)
	PCV2b	6	0	0	0
	PCV2d	6	0	1	1
	PCV2i	2	0	0	0
2017	PCV2a	0 (234)	0 (145)	0 (56)	0 (28)
	PCV2b	22	2	7	0
	PCV2d	34	0	5	0
	PCV2i	2	0	0	0
2018	PCV2a	8 (337)	0 (0)	0 (0)	0 (0)
	PCV2b	10	0	1	0
	PCV2d	12	0	0	0
	PCV2i	3	0	0	0
2019	PCV2a	0 (125)	0 (0)	0 (0]	0 (45)
	PCV2b	2	0	0	5
	PCV2d	3	0	0	4
	PCV2i	0	0	0	2
2020	PCV2a	0 (116)	0 (0)	6 (19)	0 (0)
	PCV2b	0	0	1	0
	PCV2d	1	0	4	0
	PCV2i	0	0	0	0
2021	PCV2a	5 (150)	0 (0)	0 (0)	0 (0)
	PCV2b	5	0	0	0
	PCV2d	6	0	0	0
	PCV2i	1	0	0	0

Note: The numbers in parentheses represent the number of samples.

**Table 5 animals-14-02882-t005:** Nucleotide sequence differences among nine genotypes of PCV2. This study included 251 full-length sequences of PCV2 obtained in this study and eight complete sequences representing all eight PCV2 genotypes (a–h) retrieved from GenBank.

Nt Position	PCV2a	PCV2b	PCV2c	PCV2d	PCV2e	PCV2f	PCV2g	PCV2h	PCV2i
617	C	C	C	C	C	C	C	C	T
1164	C	C	C	C	C	C	G	G	T
1596	G	G	G	G	G	G	G	G	C
1604	G	G or T	G	G	G	G	G	G	A

## Data Availability

All data, including the names of the PCV2 strains and accession number(s) presented in this study, are available in the GenBank repository.

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
