# Peer review of "Frequency and Genetic Analysis of Porcine Circovirus Type 2, Which Circulated between 2014 and 2021 in Jiangsu, China"

_animals, 2024, doi:10.3390/ani14192882_

Round 1

Reviewer 1 Report

Comments and Suggestions for Authors

The paper reports the results of the detection and genotyping of Porcine circovirus type 2 (PCV2) in biological samples from pig farms in Jiangsu Province, China. The authors observed an increase in the infection rate from 2014 to 2021 and genotypes PCV2a, PCV2b, PCV2d, and PCV2i were detected. By analyzing of the genome of PCV2i strains the authors individuated specific characteristics of this genotype.

The study presents very important data concerning PCV2i genotype, but the study design does not fulfil the purpose of epidemiological analysis. Indeed, the study raises some critical issues.

First of all, in the title and in many parts of the text (lines 113, 114, 118,159,161, 267) the authors use the term Prevalence, but the study design should had been different in order to calculate the prevalence of the PCV2 infection in the pig population in the selected area

Moreover, in the summary (line 26) and elsewhere in the text, the authors claim to have calculated the infection rate of PCV2 in pig farms in Jiangsu Province. Actually, in lines 76-80 the authors state that clinical samples were collected from pig farms in 13 cities in Jiangsu Province and that most of the pigs from these farms showed clinical signs of systemic circovirus disease also referred to as PMWS. However, it is not clear whether the sampled farms were randomly selected or only farms experiencing PCVAD were selected. Besides, it is not mentioned if the sampled animals were randomly chosen. In any case, this type of sampling cannot be regarded as correct for the purpose of verifying the frequency of infection of this virus, considering that PCV2 infection can cause even very severe clinical forms but can also be asymptomatic.

Another critical point is that the rate of positivity should also be referred to the herds, and to the number of herds sampled per city. The authors state they studied PCV2 infection in pig farms in Jiangsu Province (see lines 25-26, 33, 70 ,77 ,79 ,119, 211) but they do not mention how many farms were sampled and do not consider in their experimental design the farms of origin of the sampled animals. They mention that samples were retrieved from 13 cities (line 33, 119), but neither the information concerning the number of farms sampled from every city nor the distribution of the samples in these farms are considered.

The farm of origin of the different genomes should also be considered. The authors, at line 94 state that 251 strains were randomly selected to amplify the complete genome of the virus. Then, in lines 134-135 they state: “the 251 PCV2 strains were randomly selected from different regions among the PCV2-positive samples to investigate the genetic features of recent PCV2 strains prevalent in Jiangsu Province”. The farm of origin should not be ignored in this process. In addition, at line 139 the authors write: “The 251 PCV2 strains shared 94.1–100.0% (complete genome)”. It is crucial to know if strains with the 100% of identity come from the same herd or not. The same is applies to the different genotypes. In particular, it is essential knowing the origin of PCV2i strains.

Another aspect that is not mentioned in the study is whether or not the sampled animals had been vaccinated against PCV2. Indeed, although vaccination does not confer sterilising immunity, it is very effective in reducing the clinical forms, the duration of viremia and viral titre, and consequently PCR positivity. The authors should also consider the vaccination status of sampled herds in their study design.

There are also other issues to consider.

Lines 23-24: Please note that it is not univocally clear that new genotypes play an important role in posing significant challenges of the disease prevention and control.

See for example the review: “Porcine Circovirus 2 Genotypes, Immunity and Vaccines: Multiple Genotypes but One Single Serotype”, Pathogens 2020, 9, 1049; doi:10.3390/pathogens9121049

Lines 28-29: the assertion is not consistent with the study.  

Line 32: the subdivision in 9 serotypes is not “believed”, it is based on sequence analysis and conventional parameters.

In table 2 page 4. The authors show the infection rate of sample from the 13 sampled cities, but there is no correspondence between the names of the cities in column 1 and the 13 rows.

At line 57 the authors state the genome is “mainly divided in two open reading frames”. Please check this information and update it.

At lines 174 and 187 the term homology is not correct. Please change it.

Author Response

Reviewer 1

The paper reports the results of the detection and genotyping of Porcine circovirus type 2 (PCV2) in biological samples from pig farms in Jiangsu Province, China. The authors observed an increase in the infection rate from 2014 to 2021 and genotypes PCV2a, PCV2b, PCV2d, and PCV2i were detected. By analyzing of the genome of PCV2i strains the authors individuated specific characteristics of this genotype.

The study presents very important data concerning PCV2i genotype, but the study design does not fulfil the purpose of epidemiological analysis. Indeed, the study raises some critical issues.

First of all, in the title and in many parts of the text (lines 113, 114, 118,159,161, 267) the authors use the term Prevalence, but the study design should had been different in order to calculate the prevalence of the PCV2 infection in the pig population in the selected area

Thank you for your comments. Based on your and reviewer 3’ suggestions, we have changed the term “prevalence” to “frequency”.

Moreover, in the summary (line 26) and elsewhere in the text, the authors claim to have calculated the infection rate of PCV2 in pig farms in Jiangsu Province. Actually, in lines 76-80 the authors state that clinical samples were collected from pig farms in 13 cities in Jiangsu Province and that most of the pigs from these farms showed clinical signs of systemic circovirus disease also referred to as PMWS. However, it is not clear whether the sampled farms were randomly selected or only farms experiencing PCVAD were selected. Besides, it is not mentioned if the sampled animals were randomly chosen. In any case, this type of sampling cannot be regarded as correct for the purpose of verifying the frequency of infection of this virus, considering that PCV2 infection can cause even very severe clinical forms but can also be asymptomatic.

To avoid misunderstandings, we have revised the relevant content (line 64).

Another critical point is that the rate of positivity should also be referred to the herds, and to the number of herds sampled per city. The authors state they studied PCV2 infection in pig farms in Jiangsu Province (see lines 25-26, 33, 70 ,77 ,79 ,119, 211) but they do not mention how many farms were sampled and do not consider in their experimental design the farms of origin of the sampled animals. They mention that samples were retrieved from 13 cities (line 33, 119), but neither the information concerning the number of farms sampled from every city nor the distribution of the samples in these farms are considered.

Thank you for your suggestion. We have modified the expression of the content in Table 2 and added Table 3 to address the issue.

The farm of origin of the different genomes should also be considered. The authors, at line 94 state that 251 strains were randomly selected to amplify the complete genome of the virus. Then, in lines 134-135 they state: “the 251 PCV2 strains were randomly selected from different regions among the PCV2-positive samples to investigate the genetic features of recent PCV2 strains prevalent in Jiangsu Province”. The farm of origin should not be ignored in this process. In addition, at line 139 the authors write: “The 251 PCV2 strains shared 94.1–100.0% (complete genome)”. It is crucial to know if strains with the 100% of identity come from the same herd or not. The same is applies to the different genotypes. In particular, it is essential knowing the origin of PCV2i strains.

We have modified the expression of the content in Table 2 and added relevant content (lines 112-113, 135-138).

Another aspect that is not mentioned in the study is whether or not the sampled animals had been vaccinated against PCV2. Indeed, although vaccination does not confer sterilising immunity, it is very effective in reducing the clinical forms, the duration of viremia and viral titre, and consequently PCR positivity. The authors should also consider the vaccination status of sampled herds in their study design.

We have added relevant content to explain the vaccination status of pig farms (lines 64-65).

There are also other issues to consider.

Lines 23-24: Please note that it is not univocally clear that new genotypes play an important role in posing significant challenges of the disease prevention and control.

See for example the review: “Porcine Circovirus 2 Genotypes, Immunity and Vaccines: Multiple Genotypes but One Single Serotype”, Pathogens 2020, 9, 1049; doi:10.3390/pathogens9121049

 Thank you for your suggestion. We have revised the relevant content (line 17) and added the above-mentioned literature to this article (Reference 27).

Lines 28-29: the assertion is not consistent with the study. 

We have deleted the assertion as suggested.

Line 32: the subdivision in 9 serotypes is not “believed”, it is based on sequence analysis and conventional parameters.

Thank you for your suggestion. We have revised the relevant description (line 23).

In table 2 page 4. The authors show the infection rate of sample from the 13 sampled cities, but there is no correspondence between the names of the cities in column 1 and the 13 rows.

We have revised Table 2.

At line 57 the authors state the genome is “mainly divided in two open reading frames”. Please check this information and update it.

We have updated the relevant information (lines 43-44) and added two references (Ref.10 and 11)

At lines 174 and 187 the term homology is not correct. Please change it.

Thank you for your attention. We have revised the term “homology” (line 148, 158).

Reviewer 2 Report

Comments and Suggestions for Authors

80 When you express clinical signs of PMWS they are inespecific. Please try to be more specific as well as add some information about gross and/or microscopic lesion. Please indicate type of farms (indoors or outdoors) as well as if the farms were vaccinated against PCV-2 

Table 1. Reemplace Clinical symptoms . by suspected PMWS or Clinical signs of PMWS. 

247-253  Please try to added aditional information on PCV2i such as from wich type of farms were identified, organs and /or any type of lesions

Author Response

Reviewer 2

80 When you express clinical signs of PMWS they are inespecific. Please try to be more specific as well as add some information about gross and/or microscopic lesion. Please indicate type of farms (indoors or outdoors) as well as if the farms were vaccinated against PCV-2 

Firstly, we would like to thank you for your positive comments concerning our article. We have added the relevant information (lines 40-42, 62, and 64-65).

Table 1. Reemplace Clinical symptoms . by suspected PMWS or Clinical signs of PMWS.

 Thank you for your suggestion. We have deleted this column from Table 1 and expressed the relevant content in words (line 64). 

247-253  Please try to added aditional information on PCV2i such as from wich type of farms were identified, organs and /or any type of lesions

We have added Table 4 and the relevant content (lines 135-138) to illustrate the above issues.

Reviewer 3 Report

Comments and Suggestions for Authors

Comments

Title

Change the term prevalence to frequency.

Abstract

Lane 33, mention that the dominant genotype (PCV2d) is currently. Previously, they were genotypes a and b.

Lane 38-40, indicate the progression of detection by genotype in the years analyzed. Indicate how the detection frequency was for each genotype and if the percentage varied over time.

Introduction

Indicate in the introduction how the massive use of vaccination influenced the emergence of new genotypes and how the circulation of PCV2 has been maintained in swine populations with immunity derived from vaccination or natural infection.

Lane 65, specify the factors associated with the two changes in the dominant genotypes. It is important to point out the genetic and antigenic characteristics that have led to this shift in genotypes over time.

Lane 68, indicate in the previous study which genotypes were detected and in what proportions.

M&M

Lane 77, indicate how many farms and their distribution by city.

Lane 79, it is mentioned that most of the farms presented PMWS, but in table 1 the 2675 samples are added, correct that paragraph.

Table 1, if possible indicate the proportion or the n of each type of sample (lung, serum, stool and lymph nodes).

Lane 83-91, for molecular detection, a 630 bp fragment was amplified by endpoint PCR. What is the sensitivity of your test to be used to determine prevalence? Is a real-time PCR test or a nested PCR better? Explain.

Lane 92-99, mention that 251 positive samples were chosen at random to have the complete genome amplified. What method or statistical test did you use? Explain what your criteria was. It is difficult for it to be random.

Lane 103-112, use only 8 sequences to reference each genotype, is it representative? Explain.

Results

Lane 118-128 and Table 2, indicate the percentage of positivity according to the type of sample used. Pay special attention to the cases where no positive samples were found, could it be because there was little representativeness of samples or the type of sample analyzed?

There is not enough information to determine a prevalence, there was no sampling model, there is no data for the total population, among others. This study may be frequency, not prevalence

Lane 133-144, explain what happens in the translation of viral proteins according to the insertions or deletions identified. Identify and include in the results the modifications in the epitopes, if they exist or if not, mention that there are no changes.

In Figure 1, indicate which are the reference sequences that you used in the analysis.

Did you not create a phylogenetic tree by ORF? If you have them, you can include them as supplementary material.

Lane 159-161, detail how the genotypes were distributed according to the type of sample. Was there overrepresentation of any type of sample that favored the detection of a particular genotype?

Table 3, what changes in the corresponding aa were presented? How does that affect the protein in its structure? Or in the epitopes? Explain in the results whether or not there were alterations.

In the discussion, it indicates that they have identified a new genotype, PCV2i, this was achieved based on the phylogenetic classification methodology proposed by Franzo & Segalés. In this particular case, having a manuscript with a title that implies an epidemiological study, you must provide more information on the farms where positive cases for PCV2i were presented, since this information is of great relevance. Please complement and discuss it.

Author Response

Reviewer 3

title

Change the term prevalence to frequency.

Firstly, thanks for your professional review work on our article.

We have changed the term prevalence to frequency.

Abstract

Lane 33, mention that the dominant genotype (PCV2d) is currently. Previously, they were genotypes a and b.

We have revised this sentence as requested (line 23).

Lane 38-40, indicate the progression of detection by genotype in the years analyzed. Indicate how the detection frequency was for each genotype and if the percentage varied over time.

We have added the relevant content (line 29) as suggested.

Introduction

Indicate in the introduction how the massive use of vaccination influenced the emergence of new genotypes and how the circulation of PCV2 has been maintained in swine populations with immunity derived from vaccination or natural infection. Lane 65, specify the factors associated with the two changes in the dominant genotypes. It is important to point out the genetic and antigenic characteristics that have led to this shift in genotypes over time.

We have added the relevant content (lines 50-53) and three relevant references (Ref. 27-29).

Lane 68, indicate in the previous study which genotypes were detected and in what proportions.

We have added the relevant content as requested (lines 54-55).

M&M

Lane 77, indicate how many farms and their distribution by city.

We have added Table 3 and the relevant content (lines 99-101) as requested.

Lane 79, it is mentioned that most of the farms presented PMWS, but in table 1 the 2675 samples are added, correct that paragraph.

We have revised the content of Table 1 and related descriptions (line 64).

Table 1, if possible indicate the proportion or the n of each type of sample (lung, serum, stool and lymph nodes).

We have added Table 4 and the relevant content (lines 128-131) as requested.

Lane 83-91, for molecular detection, a 630 bp fragment was amplified by endpoint PCR. What is the sensitivity of your test to be used to determine prevalence? Is a real-time PCR test or a nested PCR better? Explain.

The literature on the sensitivity and specificity of PCR amplification of 630 bp has been published, and we have supplemented this literature (line 70, Ref.32). Compared with this PCR, real-time PCR and nested PCR are more time-consuming and expensive.

Lane 92-99, mention that 251 positive samples were chosen at random to have the complete genome amplified. What method or statistical test did you use? Explain what your criteria was. It is difficult for it to be random.

The criteria for selecting PCV2 sequencing are as follows:

  1. Positive samples from different years and cities must be selected for sequencing;
  2. Determine the number of sequencing based on approximately 20% of positive cases, and prioritize samples with high PCR band concentration;
  3. All positive samples with less than or equal to 5 will be sequenced.

Lane 103-112, use only 8 sequences to reference each genotype, is it representative? Explain.

We have added two references (line 85, Ref.16, 34) to demonstrate that these 8 sequences can represent different genotypes of PCV2.

Results

Lane 118-128 and Table 2, indicate the percentage of positivity according to the type of sample used. Pay special attention to the cases where no positive samples were found, could it be because there was little representativeness of samples or the type of sample analyzed?

We have revised the data in Table 2 using different expressions.

There is not enough information to determine a prevalence, there was no sampling model, there is no data for the total population, among others. This study may be frequency, not prevalence

Thank you for your suggestion. We have changed the term prevalence to frequency. Additionally, we have added Tables 3 and 4 to address some issues.

Lane 133-144, explain what happens in the translation of viral proteins according to the insertions or deletions identified. Identify and include in the results the modifications in the epitopes, if they exist or if not, mention that there are no changes.

We have added the relevant content (lines 113-116) and a reference (35) as requested.

In Figure 1, indicate which are the reference sequences that you used in the analysis.

We have supplemented the reference strain information in Table 1 as required (lines 125-127).

Did you not create a phylogenetic tree by ORF? If you have them, you can include them as supplementary material.

We did not create a phylogenetic tree by ORF, as the ORF2 gene can be used for genetic evolution analysis due to its high mutation rate, and the evolutionary tree constructed by ORF2 is consistent with the evolutionary tree constructed by the complete genome sequence.

Lane 159-161, detail how the genotypes were distributed according to the type of sample. Was there overrepresentation of any type of sample that favored the detection of a particular genotype?

We have supplemented Table 4 as required.

Table 3, what changes in the corresponding aa were presented? How does that affect the protein in its structure? Or in the epitopes? Explain in the results whether or not there were alterations.

We have added the relevant content (lines 163-164) and a reference (35) as requested.

In the discussion, it indicates that they have identified a new genotype, PCV2i, this was achieved based on the phylogenetic classification methodology proposed by Franzo & Segalés. In this particular case, having a manuscript with a title that implies an epidemiological study, you must provide more information on the farms where positive cases for PCV2i were presented, since this information is of great relevance. Please complement and discuss it.

Thank you for your attention. The new genotype PCV2i was not discovered by our team. In this paper, we only analyzed its genomic characteristics. We have provided information on the samples from which PCV2i originated in Table 4.

Round 2

Reviewer 1 Report

Comments and Suggestions for Authors

The authors resolved the various critical issues raised by the reviewer.

Reviewer 3 Report

Comments and Suggestions for Authors

I thank the authors for taking my comments into account; the manuscript has improved favorably.